# Preferential Treatment as a Tool for Managing the Coastal Area Sustainable Development: The Case of the Vladivostok Free Port

Roman Vladimirovich Fedorenko * and Galina Anatolievna Khmeleva

Department of World Economy, Samara State University of Economics, 443090 Samara, Russia; galina.a.khmeleva@yandex.ru
* Correspondence: fedorenko083@yandex.ru

**Abstract:** With the adoption of the sustainable development goals (SDGs), the world has recognized the need to move to responsible governance in many areas of life, including seaports, which are at the forefront of economic activity and environmental safety. The present paper examines the challenges and opportunities associated with the implementation of sustainable development principles under the free port scheme. The authors analyzed the Russian Audit Chamber report on the activities of the free port of Vladivostok and compared it with the pioneer experience of the sustainable development of the port of Antwerp. The results show that focusing only on the economic and social objectives of preferential treatment is not sufficient for the effective management of coastal areas, such as ports. To improve management efficiency and fully integrate the coastal area with preferential treatment in the world economic relations, the authors consider it necessary to ensure commitment to the goals of sustainable development and propose a model for the implementation of the sustainable development principles, as exemplified by the free port of Vladivostok.

**Keywords:** integrated management; coastal area; preferential treatment; sustainable development; port infrastructure; free port of Vladivostok

## 1. Introduction

Governments have introduced preferential treatment modes in coastal areas with a focus on addressing social and economic development issues. Preferential treatment in ports is a special regime that supports lowering taxes, introducing special investment programs and the strengthening of economic development. However, as evidenced in practice, the fact that the economic activity development entails an increase in the negative impact on the environment is not always taken into account. The United Nations World Ports Sustainability Program (WPSP) [1] calls for a change in this situation and specifies a set of tasks that can only be addressed through integrated management, as they cover a wide range, from providing alternative energy sources and sustainable infrastructure, to port operations, culture and ethics. In this paper, the authors tried to estimate the results of the preferential treatment program of the free port of Vladivostok and compared it with the pioneer experience of the sustainable development of the port of Antwerp.

In recent years, the scientific community has been raising issues of studying the issues of integrating the tasks of port areas' economic development and reducing the negative impact on the environment [2]. To ensure the sustainable development of the port industry and coastal areas, it is essential to take into account the economic, social or environmental aspects of the industry development [3].

More than 80% of the world's trade is carried out by sea. Preferential treatment attracts economic activity to the coastal area, and ports play a key role in sustainable development and prosperity. Ports play an important role in international trade and contribute to the economic growth and development of regions and countries [4]. The active exploitation of

maritime transport is both a source of economic prosperity for coastal regions and a threat to their sustainable development [5].

The COVID-19 crisis is expected to act as a catalyst for active action to achieve the sustainable development goals (SDGs) in coastal areas. To this extent, the SDGs are excellent benchmarks for the transition to responsible entrepreneurship.

Seaports are the most important participants in the global transportation system, and they have always attracted the attention of researchers.

In this paper, the authors aim to contribute to the existing literature to understand how port development projects impact coastal areas and offer their own perspective on conflict resolution between various activities in order to minimize their negative economic consequences through trade-offs and sustainable development. The research object is the free port of Vladivostok (Russia) and the port of Antwerp (Belgium). The free port of Vladivostok is the largest maritime infrastructure in the Primorsky Krai of Russia and the Eastern Arctic.

The authors provide arguments in favor of the wider use of preferential treatment for sustainable development and the integration of coastal areas into global economic relations and propose a model of transition to the sustainable development principles, as exemplified by the free port of Vladivostok.

The authors claim that preferential treatment can make a significant contribution to the "blue economy" and the sustainability of regional development in the Far East and the Arctic.

## 2. Research Area

Under the federal law, the free port of Vladivostok is one of the preferential terms areas. It was introduced in 2015 in order to expand cross-border trade, develop transport infrastructure and include Primorsky Krai in the global transport network. An important aim also was to attract investment, create a network of logistics centers with special terms of transportation, storage and partial processing of goods, organize non-resource export-oriented manufacturing and increase production with high added value [6].

The free port regime of the port of Vladivostok is expanding. It has been established on the territory of 16 municipalities and forms a seaports cluster, extending to the key harbors of the Far East-Vladivostok, Pevek, Petropavlovsk-Kamchatsky, Vanino, and Korsakov. Table 1 contains basic data of the free port of Vladivostok.

**Table 1.** Basic data of the free port of Vladivostok (as of 1 July 2020).

| Ports | Number of Resident Companies | | Estimated Investment | | Number of Jobs Created | | Investing Countries |
|---|---|---|---|---|---|---|---|
| | ea. | % | Mln Euros * | % | ea. | % | |
| Vladivostok | 1716 | 88.0 | 9085 | 82.4 | 83,093 | 90.1 | Russia, China, Japan, South Korea, Taiwan, Lithuania, Hong Kong, Singapore, England, India, Vietnam, USA, Canada, Germany, the Democratic People's Republic of Korea (DPRK) |
| Pevek | 8 | 0.4 | 625 | 0.1 | 399 | 0.4 | |
| Petropavlovsk-Kamchatsky | 155 | 7.9 | 146 | 1.3 | 2399 | 2.6 | |
| Vanino | 22 | 1.1 | 1490 | 13.5 | 4125 | 4.5 | |
| Korsakov | 50 | 2.6 | 293 | 2.7 | 2197 | 2.4 | |
| Total | 1951 | 100.0 | 11,639 | 100 | 92,213 | 1000 | |

* Estimated investments in euros were calculated by authors using the euro to Russian ruble exchange rate equal to 87.99. Source: compiled by the authors based on data [7].

The center of the free port of Vladivostok cluster is the port of Vladivostok, where 88% of resident companies are concentrated and it is expected to attract 799.4 billion rubles (equal to 9.08 billion euros) of investment and create more than 80 thousand jobs. Terrotry of the free port of Vladovostok is presented in Figure 1.

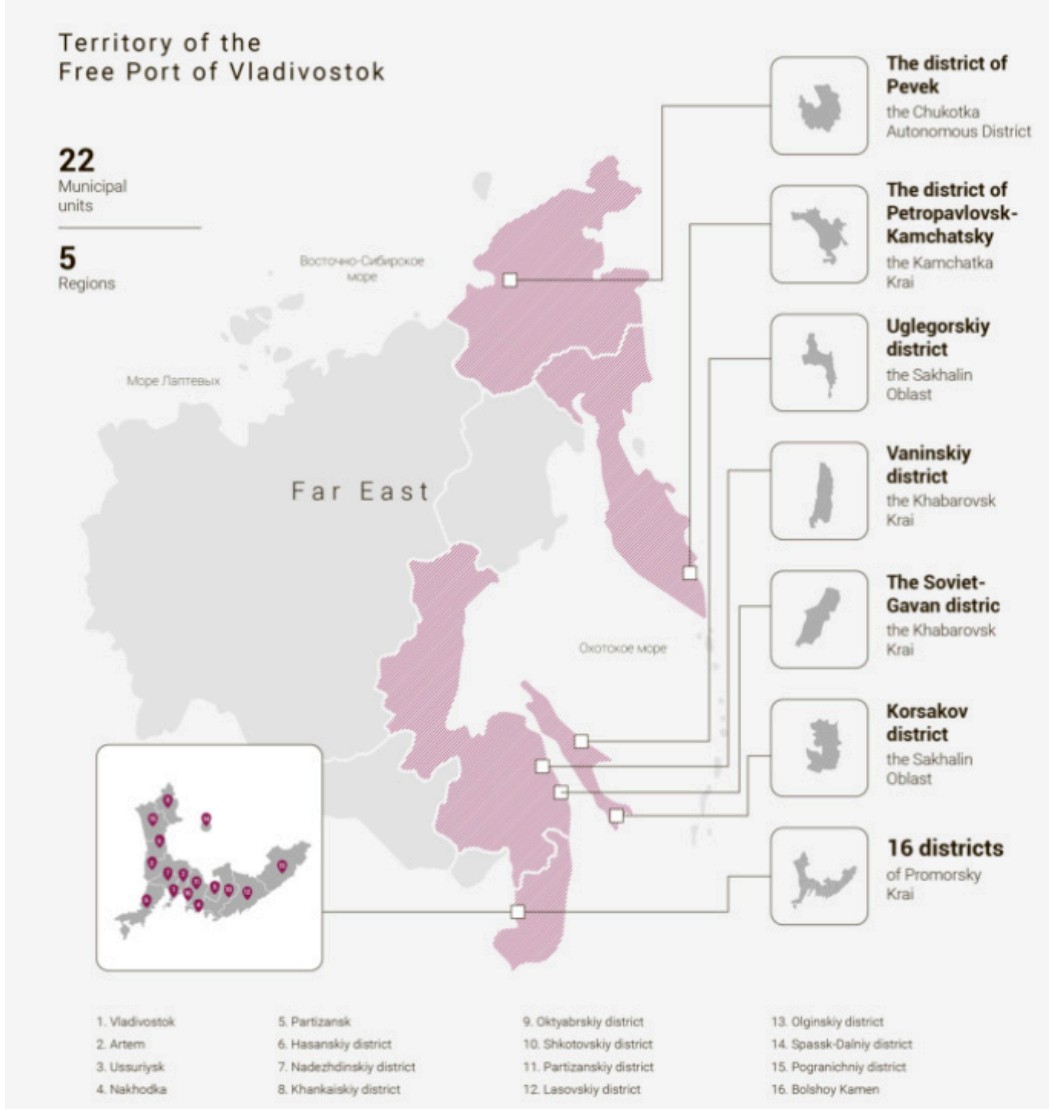

**Figure 1.** Territory of the free port of Vladivostok. Source: official site of Far East Development Corporation. https: //erdc.ru/en/about-spv/ (accessed on 21 January 2021).

Vladivostok has always been a strategically important Far Eastern outpost and growth point due to its access to the rich natural resources of the "backland" (vast spaces of Siberia and the Far East) and the equally generous "forland" (promising markets of the Asia–Pacific region). In the history of the port of Vladivostok, attempts to introduce the free port regime were made repeatedly in the 19th century and the beginning of the 20th century, but they did not bring significant benefits. Currently, the introduction of the free port regime is designed to accelerate Russia's integration into the economy of the Asia–Pacific region [8,9]. In 2019, the share of seaports in the Far Eastern basin reached more than 25% of the Russian cargo turnover [10]. The free port of Vladivostok area of operation includes large promising international transport corridors "Primorye-1" and "Primorye-2", the launch of which will have a significant impact on the regional economy by ensuring the transit of goods from the north-eastern provinces of China to the ports of Primorye, with the subsequent shipment to vessels to the countries of the Asia–Pacific region.

The direct proximity to the Arctic zone provides additional transport opportunities. It is no accident that the port city of Pevek is included in the free port of Vladivostok. Significant climate changes create longer navigation windows, as the researchers state that

the total number of passages has been steadily increasing by an average of 5506 unique vessels per year [11].

The increase in economic and transport activity will inevitably contribute to the escalation of conflict between economic interests and environmental sustainability in coastal areas. However, the management system of the Vladivostok free zone so far focuses only on achieving economic and social goals.

## 3. Methods

The first stage of the study was an in-depth analysis of national, regional and local documents related to the preferential treatment of the free port of Vladivostok (Table 2). The analysis made it possible to identify the weaknesses of management.

**Table 2.** The list of analyzed international, national, regional and local documents.

| Receiving Body | Information Type | Reference |
| --- | --- | --- |
| State Duma of the Russian Federation | About the free port of Vladivostok | SDRF, 2015 [6] |
| Government of the Russian Federation | Spatial development strategy of the Russian Federation for the period of up to 2025 | GVRF, 2019 [12] |
| Government of the Russian Federation | Socio-economic development program of the Far East and Baikal region | GVRF, 2014 [13] |
| Government of the Russian Federation | Supervisory Board composition of the free port of Vladivostok | GVRF, 2015 [14] |
| Government of the Russian Federation | Selection criteria for resident companies of the free port of Vladivostok | GVRF, 2015 [15] |
| Ministry of Regional Development of the Russian Federation | Integrated coastal area management model in Arctic regions | Ministry of Regional Development of the Russian Federation, 2014 [16] |
| Government of Primorsky Krai | Territorial scheme of waste management in the Primorsky Krai, including solid municipal waste. | GPK, 2019 [17] |

The authors provide the financial performance of the Russian port of Vladivostok using statistics in rubles. To make the numbers clear, they are also given in euros according to the exchange rate of 87.99 rubles per 1 euro (March 2021).

In the second stage, the authors studied the experience of the seaport of Antwerp in the field of sustainable development. We chose this port because, following a leadership strategy, the port of Antwerp is a pioneer in the transition to a sustainable operating model [18]. This helped to understand how it is possible to gain additional advantage from the preferential treatment in a modern context, properly taking into account the interests of business, the state and local communities.

At the final stage, the authors proposed a model for implementing the sustainable development principles in the activities of the free port of Vladivostok.

## 4. Results

### 4.1. Development Dynamics of the Free Port of Vladivostok

The favorable preferential treatment has contributed to a rapid increase in the number of resident companies of the free port of Vladivostok, including those who arrived from other regions. Since 2015, the number of resident companies has been steadily increasing from year to year and by the end of 2020 amounted to 2264 companies.

Even today, the results of preferential zones have exceeded expectations in terms of attracted investments volume. In 2019, a number of resident companies entered the construction and installation stage. 53.3 billion rubles (equal to 0.6 billion euros) were allocated to resident companies of the free port of Vladivostok allocated for these purposes [19]. Although the industrial production index from January–November 2020 in the Primorsky

Krai as a whole was 83%, there was an increase in investment in the free port of Vladivostok by 34% [20].

The industry profile of resident companies is quite diverse, given that the preferential treatment is applied to a fairly large territory of 22 municipalities and mild restrictions on the privileges application by industry. Therefore, enterprises in the field of fishing, manufacturing, mining (prohibition of oil and gas production), transportation and storage, construction and services operate here. Enterprises independently provide themselves with the necessary infrastructure. Land plots are purchased in accordance with the general rules at auction. Nevertheless, tax, customs, and administrative preferences are attractive for starting a business in the coastal marine areas of the Far East and the Arctic.

An increase in the number of resident companies, their business, entrepreneurial and export–import activity will inevitably impact not only the economy, but also the ecology of the region. Today, the highly urbanized area of the evolving Vladivostok agglomeration, with large and big cities, a significant share of the urban population, housing and service facilities, produces many times more volume and growth of waste, domestic and industrial, than any other territory outside of the agglomeration areas [17]. Marine terminal operators, together with coal and mining companies and enterprises of the heat and power sector, generate more than 90% of the total amount of waste in the analyzed coastal areas.

### 4.2. Analysis of the Preferential Treatment Application Effect in the Free Port of Vladivostok

In 2020, the Accounts Chamber of the Russian Federation presented the results of an analysis of the free port of Vladivostok effectiveness, based on the compliance of performance results reported for 2017–2018 to the goals set in the official regulatory and legislative acts.

The free port of Vladivostok was established with the purpose to develop international trade with countries in the Asia–Pacific region (APR). The aim was to create and develop industries based on modern technologies application and oriented at producing competitiveness in the APR countries products, as well as accelerating the socio-economic development of the free port of Vladivostok territory and improving the quality of life of the Far East population [6].

Preferential treatment involves the establishment of a better (competitive) environment for business and investment activities as compared to similar territories [21]. Due to this, the core documents on strategic development consider the free port of Vladivostok as an instrument of advanced economic development [12,13].

Initially, the infrastructure construction was intended as a public–private partnership. However, the public money of the Russian Federation budgetary system is not used for the creation and operation of the free port of Vladivostok.

During 2017–2018, 743.8 tons of imported goods worth 544 million rubles (equal to 6.2 million euros) were imported into the territory of the free port of Vladivostok and put under the customs procedure of the free customs zone. The total amount of privileges provided for the payment of customs duties and taxes amounted to 56 million rubles (0.6 million euros). The largest volume of goods by weight and value was imported in 2017 and amounted to 471.6 tons, worth 493 million rubles (equal to 5.6 million euros), or 74.05% of the weight and 90.6% of the value of all goods imported into the territory of the free port of Vladivostok. The privileges provided for the payment of customs duties and taxes amounted to 42 million rubles (equal to 477 thousand euros). Table 3 shows data on the volume and value of goods imported into the territory of the free port of Vladivostok and put under the customs procedure of the free customs zone, and the amounts of privileges provided for the payment of customs duties and taxes [22].

The goods imported into the territory of the free port of Vladivostok in 2017 and placed under the customs procedure of the free customs zone are primarily industrial production equipment, furniture, lamps and lighting equipment, the share of which is 86%. In 2018, customs duties and taxes were paid in the amount of 3.38 million rubles (equal to 38.5 thousand euros). During the two years, 3.7 million rubles (equal to 42.5

thousand euros) were paid as duties and taxes at placing goods under the final customs procedures [22].

**Table 3.** Data on the cost of goods imported into the territory of the free port of Vladivostok and privileges from 2017–2018 *.

| Years | Net Weight | Cost of Goods | The Customs Duty Privileges Amount | The VAT Privileges Amount | The Customs Duties and Taxes Privileges Amount | Share of Total Weight | Share of Total Cost |
|---|---|---|---|---|---|---|---|
| | kg | Thousands of Euros | Euros | Thousands of Euros | Thousands of Euros | % | % |
| 2017 | 471,591.85 | 5603.5 | 73.81 | 409.01 | 482.82 | 63.40 | 90.58 |
| 2018 | 272,203.46 | 582.77 | 40.28 | 119.86 | 160.15 | 36.60 | 9.42 |

* Estimated investments in euros were calculated by authors using the euro to Russian ruble exchange rate equal to 87.99. Source: [22].

During the investment projects' implementation in the free port of Vladivostok, resident companies paid 6.4 billion rubles (excluding VAT) of taxes, contributions, and customs duties to the budgets of the Russian Federation budgetary system (equal to 72.7 million euros). However, the free port of Vladivostok operation has not yet led to an achievement of significant results in terms of the socio-economic development of the proper constituent entities of the Russian Federation.

The total amount of lost income of Russian Federation entities budgets, on the territory of which the free port of Vladivostok operates, amounted to 445.3 million rubles (equal to 5.1 million euros) in 2017, and 857.9 million rubles (equal to 9.75 million euros) in 2018 [22].

The migration outflow of the population from the Primorsky Krai from 2016–2018 amounted to 13.5 thousand people, and from the Khabarovsk Krai, 10.2 thousand people. The total amount of public debt of the Primorsky Krai increased from 4.4 billion rubles in 2017 to 5.2 billion rubles in 2018 (equal to 50 million euros in 2017 and 59 million euros in 2018), and of the Khabarovsk Krai, public debt increased from 41.1 billion rubles in 2017 to 49.1 billion rubles in 2018 (equal to 467 million euros in 2017 and 558 million euros in 2018). The employers' needs for employees declared to the employment services increased in 2018 by 34.6% in the Kamchatka Krai, by 7% in the Primorsky Krai and by 34.5% in the Chukotka autonomous districts. At the same time, the unemployment rate in the Kamchatka Krai increased by 15% in 2018 [22].

### 4.3. The Port of Antwerp Experience of Progressing towards the SDGs

#### The Port of Antwerp: Introduction

The port of Antwerp, acting as a major hub and gateway for international trade, is of great interest to researchers as a participant in fierce competition between seaports [23], an emerging green port [24] and an example of effective management [25] and a leader in sustainable development strategy implementation.

Thanks to a convenient combination of delivery routes, well-organized work and the modern service technologies application, the port of Antwerp is one of the largest European transport hubs and has the capacity to handle cargo of almost any type and size. In 2017, the port of Antwerp provided 4.8% of Belgium's GDP, and 8% of Flanders' GDP. Moreover, port manufacturing and storage companies, whose turnover in 2017 was 5.9 billion euros, made the largest contribution. The combination of industrial production, cargo handling and high-quality logistics have ensured the synergistic effect of all activities. The Port Authority is aiming to expand container capacity to 7 million TEU (Twenty-foot equivalent unit), and is striving to achieve this through innovation, digitalization and automation.

#### The Port of Antwerp: Sustainable Development Results

In an environment where the resources for expanding the port's hinterland are almost exhausted [26], greening allows the port of Antwerp, like other ports around the world, not only to protect its license and meet government requirements for environmental safety, but also ensure economic competitiveness [17]. Therefore, the development priorities are

the transition to a circular and low-carbon economy. Table 4 presents the main indicators of sustainable development of the port of Antwerp.

**Table 4.** The main indicators of sustainable development of the port of Antwerp.

| Indicators | 2010 | 2013 | 2016 | 2018 |
|---|---|---|---|---|
| N° of calls by seagoing ships, including | 14,783 | 14,340 | 14,500 | 14,595 |
| Ships 10,000–13,000 TEU | 39 | 81 | 78 | 212 |
| Ships > 13,000 TEU | 52 | 114 | 432 | 410 |
| Degree of congestion on the main roads of Antwerp, in moving average of number of km∗h per working day | no data | 164.8 | 232.82 | 270.8 |
| Estimated emissions of nitrogen oxides (NOx) by the energy, refinery and industrial sectors in the port of Antwerp, tons | 12.8 | no data | 12.7 | 11.0 |
| Annual quantities of ship's waste and cargo residues delivered in the port of Antwerp by seagoing vessels, $m^3$ | no data | no data | 121,346.83 | 146,201 |

Source: compiled by the authors based on data [27,28].

The port of Antwerp is increasing its transportation by large-tonnage vessels, thereby changing its transportation structure. This is due to the global upward trend in the global demand for liner shipping, which is currently characterized by larger ship sizes, wide geographical coverage and frequent restructuring of shipping lines [29,30].

To prevent environmental pollution, the port uses the "polluter pays" principle and collects ship waste and cargo residues. The volume of collected residues from sea shipping, consisting of waste, household water, chemicals and oily waste, increased by 3% in 2017 compared to 2016, in 2018 by another 14% and amounted to 146,201 $m^3$.

The introduction of stricter standards at the Flemish level has helped to reduce emissions from the energy sector. If at the pan-European level, the average annual concentration of PM10 is 40 $\mu g/m^3$, then in the port of Antwerp this indicator from 2015–2017 does not exceed 25 $\mu g/m^3$ [28].

*Sustainable Development Model: Port Community and Projects*

A sustainable port implies the alignment of economic, social and environmental interests [23]. The port of Antwerp's sustainable development policy is based on five SDGs: health and well-being, work and economic growth, innovation, sustainable cities and communities, and climate action.

The sustainable development policy uses the "owner–port" model of successful cooperation. The main idea is to create the port community instead of individual companies doing business in the port. This allows the building of interaction and intensive cooperation between companies, local authorities and communities, national and global partners within the framework of a common vision of the local, regional and global context within the framework of "Vision 2030–2050" [31].

The port of Antwerp is developing a business ecosystem of a circular economy, which already provides a synergistic effect from a well-developed logistics infrastructure in the input–output chain for the purchase and sale of goods and services within the ecosystem [32], and eventually scientists register a synergistic effect of industrial ecology, i.e., the exchange of by-products through a special infrastructure. For example, companies located in port areas use each other's residual energy and chemical effluents as raw materials for their own production processes [33]. At the same time, the port of Antwerp takes an active role in achieving the synergy of industrial ecology through investments in special infrastructure and active partnerships [34]. Collaboration decreases the costs of transport services, optimizes logistics and reduces $CO_2$ emissions [35].

Providing long-term guidelines, the SDGs serve as the basis for building partnerships. In the context of the SDGs, the port of Antwerp is implementing a number of initiatives.

Thus, a number of partner projects are being implemented for sustainable growth. For example, on the former Opel site, a landfill was created for the circular industry business



implementation—the NextGen District. The port of Antwerp attracts industrial projects, in which potential waste serves as products for new use in the value chain.

At the end of 2017, the Zero Pellet Loss project was launched—an initiative for environmental control. To this end, the port of Antwerp joined the Operation Clean Sweep© international program, having brought together fifteen port companies in the framework of joining the action program to prevent the loss of pellets [36].

To increase the commodity flow sustainability, the beyond chocolate partnership project is being implemented, whose participants are the chocolate companies along with the port of Antwerp. The participating companies demonstrate their commitment to ensuring fair wages for cocoa farmers, preventing child labor and deforestation. Realizing the strategic importance of commitment to sustainability, the largest retail networks of Belgium, as well as universities, governmental and non-governmental organizations, development organizations and trade unions are involved in this effort.

The port of Antwerp is implementing other proposals as well, which, in partnership with businesses, government and citizens, enable the progressive advance towards achieving the sustainable development goals, while ensuring economic, social and environmental effects.

Partners from other regions are involved in joint activities. For this purpose, the STHIL project is being implemented to identify the residual flows of companies and sectors that may be involved in the value chains of neighboring regions. The logistics companies and the Antwerp chemical cluster have a particularly high potential for this. For example, a new opportunity is that the chemical cluster could become a major consumer of recycled components that come from waste streams elsewhere in Flanders [37].

An example of a successful global collaboration is the Antwerp@C project in a consortium with leading chemical and energy companies: Air Liquide, BASF, Borealis, Exxon-Mobil, INEOS, Fluxys, the port of Antwerp and Total. The project aims to achieve climate targets and aims to halve the port's $CO_2$ emissions (18.65 million tons of greenhouse gas emissions in 2017) by 2030. In 2020, the project received a grant of EUR 9 million from the Connecting Europe Facility (CEF). This approach allows you to get a new effect of "ecologies of scale" [38].

*The Challenges on the Road to the Port of Antwerp Sustainable Development*

In the pre-pandemic period, business activity steadily increased and the problem of a lack of suitable industrial operators was on the agenda. To mitigate the impact of this problem and contribute to the balanced and sustainable development of Belgium, the Port Authority of Antwerp is involved in joint projects with enterprises and government agencies outside the port area. For example, the Port Authority of Antwerp has initiated an interregional project 'Haven en Hinter', covering the regions of Antwerp, Waasland, Flemish-Brabant and the Kempen. The project is aimed at the efficient use of industrial and logistics areas through a coordinated policy of spatial distribution and business interaction [39].

Every year, 40 thousand schoolchildren and students not only from Belgium, but also from many countries of the world, visit the Talent Center in the port of Antwerp. Despite this, it has not yet been possible to solve the personnel problem of attracting young, qualified specialists. This problem is especially relevant for the port and logistics sectors. Due to the long-term structural shortage of personnel in the Belgian economy, it is difficult to find well-trained professionals capable of efficiently managing the growing flow of goods [40]. Along with the institutional limitations of the labor relations regulation system, the role of competition between employers for qualified personnel and supranational factors is high [41]. The processes of globalization, digitalization, and technological development require appropriate qualifications, which potential employees do not always possess [42].

Monitoring allows one not only to assess the progress of sustainable development, but also to create "new discursive spaces for concern and mobilization" [43]. The port of Antwerp, with key partners, has been publicly posting a sustainability report since 2012. All members of the port of Antwerp community can participate in its preparation and

assessment. The Antwerp Sustainability Award has been established to reward the most active contributors for innovative contributions to a sustainable port.

There is some progress in sustainable development, but it remains slow. Recent studies show that the distance from awareness to action in the Antwerp port business community is still long. However, more often this is due to the complexity and the need to coordinate the participants in real initiatives and corporate "business cases", which do not always have tangible potential effects. So far, companies are more likely to make decisions in terms of short-term survival [44].

Thus, the Antwerp model, while not devoid of flaws, offers a value model. This model is a generator of economic benefits, and takes care of environmental safety. Here, as in the case of Vladivostok, the city lands are managed with the involvement of state resources. The state, local community and business are strengthening the commitment to cooperation and collaboration. This enables progress towards the sustainable coastal development goals. Therefore, the authors propose to apply the Antwerp model in the activities of the free port of Vladivostok.

*4.4. Model of Sustainable Development Principles Implementation in the Free Port of Vladivostok Operations in the System of Integrated Coastal Area Management*

To use experience of the port of Antwerp's sustainable development, we should compare its main indicators with the free port of Vladivostok characteristics. Table 5 shows the results of a comparison based on the data of 2018.

**Table 5.** Comparison between the port of Antwerp and the free port of Vladivostok characteristics in 2018.

| Ports | Net Weight | Container Traffic | N° of Calls by Seagoing Ships | Number of Jobs Created | Estimated Investment |
|---|---|---|---|---|---|
| | Million Tons | Thousands of TEU | ea. | ea. | Billion Euros * |
| Antwerp | 235.3 | 11,000 | 14,595 | 140,000 | 3.4 |
| Vladivostok | 10.4 | 551 | 1989 | 80,000 | 1.8 |
| Ratio of indicators, Antwerp/Vladivostok | 22.62 | 19.96 | 7.34 | 1.75 | 1.9 |

* Estimated investments in euros were calculated by authors using the euro to Russian ruble exchange rate equal to 87.99. Source: compiled by the authors based on data [7,28].

The port of Antwerp handles more than 20 times higher volumes of cargo than the free port of Vladivostok. To successfully handle this amount of cargo, the port of Antwerp has about 140,000 jobs and attracts investments of about 1.8 billion euros annually. These indicators are less than two times higher than those of the free port of Vladivostok. According to the adopted program for the development of the free port of Vladivostok from 2015–2019, the estimated investment was about 799.4 billion rubles (equal to 9.08 billion euros). In 2018, the volume of investment was about 1.8 billion euros. This made it possible to increase the volume of cargo turnover by 39%. In 2019, cargo turnover increased by another 10.5% and reached a new all-time high of 11.7 million tons [45]. The port's bandwidth is still not fully utilized. The Ministry of Transport of the Russian Federation determines the estimated throughput of the free port of Vladivostok is equal to 31 million tons per year [46]. This is almost three times the record figure of 2019. As we can see from Table 4, the labor and investment resources of the free port of Vladivostok are comparable to those of the port of Antwerp. Thus, the authors suggest that it is possible to successfully implement the management experience of the port of Antwerp to ensure the sustainable development of the free port of Vladivostok.

The management complexity is evident in the broad context of participants from a variety of business dimensions: fishing and fish processing, shipping, mining, construction, health care and other services. Representatives of these areas run businesses as resident companies of the free port of Vladivostok.

The free port of Vladivostok is managed by the Supervisory Board, the authorized federal body, and the public council. The Supervisory Board consists of representatives of the government, regulatory organizations (taxes, environmental, sanitary, veterinary inspection, etc.), the minister of the relevant Ministry for the Development of the Far East and the Arctic, the heads of municipalities located on the free port of Vladivostok territory, as well as representatives of public business associations.

Currently, the basic criteria for the resident companies' selection of the free port of Vladivostok are: a new investment project or a new type of activity, and an investment of at least 5 million rubles over 3 years. The free port of Vladivostok is managed by the management company JSC, "Corporation for the Development of the Far East".

Taking into account the free port of Vladivostok management structure and successful experience of the port of Antwerp, the proposed model for implementing sustainable development in the territory of the free port of Vladivostok is presented in Figure 2.

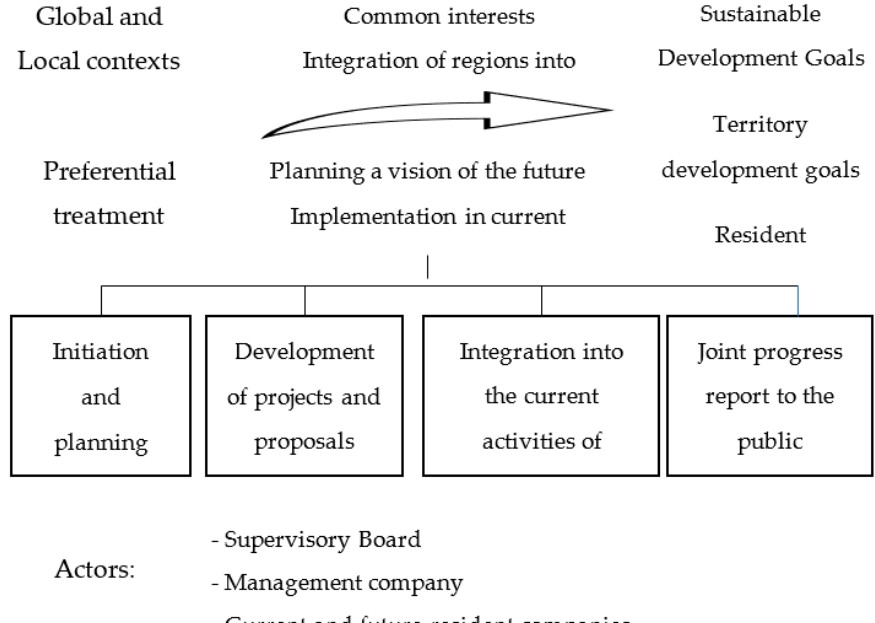

**Figure 2.** Model of the sustainable development goals (SDGs) implementation in a preferential area of the free port of Vladivostok. Source: developed by the authors.

Below are the basic aspects of presented model for the free port of Vladivostok, in more detail.

1. Initiating the program development for the sustainable development goals implementation in the territory of the free port of Vladivostok on the management company part, obtaining support from the Supervisory Board. This is essential, as the Supervisory Board, within its powers, will promote the involvement of a wide range of participants—representatives of state bodies and municipalities—in sustainable development projects.

2. Defining sustainable development goals that are relevant to the goals of the free port of Vladivostok. These goals are determined by the Supervisory Board and the management company, based on the development priorities of the free port of Vladivostok, taking into account the positive and negative, current and potential impact on the SDGs within the value chain.

3. Developing projects and proposals for the SDGs implementation in order to define a set of tools that, through partnership and cooperation, will bring economic, social and environmental benefits.

4. Sustainable initiatives integration with businesses of the free port of Vladivostok resident companies. To accomplish this, a partnership agreement with resident companies,

providing for the commitment to join the free port of Vladivostok initiatives in the field of sustainable development, should be proposed.

5. Providing joint reporting on progress towards the SDGs, using common indicators and shared priorities. This step is necessary for building effective communications between the management company, resident companies, government agencies and the public.

The proposed mechanism for achieving the common SDGs will increase the free port of Vladivostok's efficiency by attracting green investments and increasing the socio-economic and environmental impact. Moreover, the SDGs implementation increases the degree of integration of certain regions into global economic relations through practical actions in the interests of global sustainable development.

## 5. Discussion

The marine environment degradation resulting from land-based activities is an essential problem. To address this issue, the European Commission adopted a new initiative on marine spatial planning and integrated coastal area management [47].

Integrated coastal area management is an effective approach proven by international experience, as it aims to increase the economic benefits of coastal zone exploitation by minimizing the environmental impact of economic activities, as well as to support and improve social security through new jobs and income. Commitment to the SDGs is a necessary link in building an integrated coastal management system. This approach provides long-term benefits to all stakeholders. The state gains an increase in business activity, tax collection, employment growth and sustainable economic growth, and the population has a healthy environment. Commitment to the SDGs enables businesses to form sustainable strategies for long-term development based on innovation and collaboration.

The authors conclude that Russia has formed the prerequisites for the SDGs implementation in the free port of Vladivostok operations. First of all, because Russia joined the SDGs in 2019, the Russian National Climate Plan was adopted [48]. The measures developed by the Government of the Russian Federation aimed at achieving the SDGs are integrated with national projects, strategies and programs. Thus, 12 national projects and a comprehensive plan for the trunk infrastructure modernization and expansion affected directly or indirectly, 107 of 169 tasks set in the UN document [49]. A national set of the Sustainable Development Goals (SDGs) indicators has been defined, allowing for the monitoring and control of the sustainable development goals achievement at the level of Russia. It can be said that Russia has laid the foundation for systematic work to achieve the SDGs. However, serious government measures to support the transition to sustainable coastal development are still insufficient. Moreover, insufficient awareness of local communities and local governments about sustainability management is an issue. Meanwhile, the introduction of the sustainable development implementation model in the operations of the free port of Vladivostok requires reflection and serious motivation.

The issue of port infrastructure sustainable development can be considered from various perspectives. The authors focused on the advanced administrative mechanisms implementation for managing the port region development.

Modern researchers may select other elements of ensuring sustainable development as a separate aspect of the study. Serious attention is routinely paid to the issue of infrastructure support for the port regions' sustainable development. Some essential elements of many seaports' infrastructure are significantly deteriorated and stay operational for much longer than their original design lifetime [50].

Researchers pay serious attention to the complexity of environmental safety issues in combination with the active sea routes operation and coastal infrastructure. A number of modern authors highlight the importance of environmental management as a sustainable development indicator of port regions [51].

A large number of modern studies explore the issue of sustainable development [52–54]. Some researchers, as well as the authors of this paper, report the impact that coastal areas'

sustainable development has on the economic potential of adjacent regions [55], certain sectors of economy [56] and the entire country as a whole [57].

## 6. Research Limitations and Future Research

This study has its limitations. In particular, we focused on studying the preferential treatment as exemplified by the free port of Vladivostok and the seaport of Antwerp. We did not have sufficient data available on other free seaports, neither did we have more comprehensive statistical data on the free port of Vladivostok operation available in open sources. We hope that the interest in "blue economy" and the sustainable development of preferential zones will contribute to the emergence of international databases on the operation of seaports with preferential treatment. Opportunities for future research are related to quantitative measurements of the explicit and implicit costs and benefits of all participants in the preferential zone of coastal areas, the impact on certain industries' development and the relationship between them. Since preferential zone management, such as a free port, involves many participants, the study of mechanisms and results of cooperation in the joint projects implementation, depending on the participant type and participation form, would be beneficial. It is equally important to follow the resident companies' integration process at the level of the overall SDG strategy and overall infrastructure.

## 7. Conclusions

The proposed model is an attempt to conceptualize the SDGs as an element of integrated coastal preferential zone management in view of current discussions in the field of preferential zone management science and sustainable development management science.

This paper touches upon the issue of involving beneficiaries and local communities in the sustainable management of coastal preferential zones. The situation when the preferences established on the territory serve as a way to achieve only the state economic and social goals, decreases the management efficiency and leads to concealed losses for the state, resident companies and the local community, since the impact of infrastructure shared use and environmental losses from economic activity are not taken into account. Considering the global environment in the long-term development of preferential zones leads to a deeper integration of coastal areas into the global context through commitment to the shared goals of sustainable development. The authors conclude that preferential treatment should be applied not only as a way to obtain benefits of lower costs for businesses, but also for additional jobs for resident companies and taxes in the immediate future for the state. To do this, the authors proposed a model for sustainable development implementation for the coastal region preferential area and elaborated on it, as exemplified by the free port of Vladivostok; the model allows one to solve the problem of interests imbalance and lays the foundation for long-term sustainable development and mutually beneficial cooperation in the coastal region.

**Author Contributions:** Conceptualization, R.V.F.; Investigation, G.A.K.; Project administration, R.V.F.; Writing—original draft, R.V.F. and G.A.K.; Writing—review & editing, R.V.F. All authors have read and agreed to the published version of the manuscript.

**Funding:** The research was carried out with the financial support of the RFBR in the framework of research project No. 19-510-23001.

**Institutional Review Board Statement:** Not applicable.

**Informed Consent Statement:** Not applicable.

**Conflicts of Interest:** The authors declare no conflict of interest.

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
