# Peer review of "Preferential Treatment as a Tool for Managing the Coastal Area Sustainable Development: The Case of the Vladivostok Free Port"

_jmse, doi:10.3390/jmse9030329_

Round 1

Reviewer 1 Report

This is basically quite a good paper. It is logical, largely well-structured and informative. There are a few shortcomings which, if corrected, would greatly improve the quality of the paper.

The work needs to present a better geographical context. This is probably best achieved through the inclusion of a map, which is probably best incorporated into section 2, so that the boundaries of the port can be shown, the connecting road network links, local towns and proximity to the Arctic region.

Subjective statements such as ’we believe’ (at end of introduction) and 'in our opinion' (later in the paper) should be avoided. Indeed, it is good academic practice to avoid the use of all personal pronouns and statements made in the text should be backed up by evidence (references or conclusions drawn from analysis).

There needs to be an earlier and clearer definition of the nature of a free port in Russia. Perhaps the wording from section 4.2 could be incorporated into the introduction, rather than so much later in the text: "Preferential treatment involves the establishment of better (competitive) environment for business and investment activities as compared to similar territories [21]. Due to this, the core documents on strategic development consider the Free port of Vladivostok as an instrument of advanced economic development [12,13]."

The following statement is not appropriate for the main text and the exchange rate should be a note to the table it refers to: "We provide the financial performance of the Russian port of Vladivostok using statistics in rubles. Euro to Russian Ruble Exchange Rate is at a current level of 89.05 (18.02.2021)". The use of 'we' should be avoided and the exchange rate made more obvious.

In section 4.1, 'residents' can be interpreted as people or population, when clearly what is meant is companies or businesses. The text should always use the expression ‘resident companies’ rather than just residents, particularly since later in the text, populations (residents) are referred to.

The numerical figure in the following sentences is extremely confusing to a western reader.” The total amount of privileges provided for the payment of customs duties and taxes amounted to 56,576. 62 thousand rubles.”, “The privileges provided for the payment of customs duties and taxes amounted to 42,483. 79 thousand rubles”. The figures after the decimal point are trivia´l and should be deleted and the whole number figure should be quoted as either 'million' or and extra three zeros 'i.e. 000' should suffix each of these numbers. 

Section 4.3 is structurally unsound. The section on the port of Antwerp governance and regulation is largely irrelevant to the central discussion and should be deleted. The section comparing Antwerp and Vladivostok should be moved to the beginning of section 4.4. as a form of introduction to the situation in Vladivostok in comparison to Antwerp. Section 4.3 then becomes only about Antwerp.

Finally, the level of English in the paper could be improved. The paper is certainly very readable at the moment, but the English is not perfect. This should be a matter of journal policy as to whether it needs to be improved or otherwise.

Author Response

Point 1

The work needs to present a better geographical context. This is probably best achieved through the inclusion of a map, which is probably best incorporated into section 2, so that the boundaries of the port can be shown, the connecting road network links, local towns and proximity to the Arctic region.

Point 1

We added a map of the Territory of the Free Port of Vladivostok.

Point 2

Subjective statements such as ’we believe’ (at end of introduction) and 'in our opinion' (later in the paper) should be avoided. Indeed, it is good academic practice to avoid the use of all personal pronouns and statements made in the text should be backed up by evidence (references or conclusions drawn from analysis).

Point 2

Subjective statements were cganged.

Point 3

There needs to be an earlier and clearer definition of the nature of a free port in Russia. Perhaps the wording from section 4.2 could be incorporated into the introduction, rather than so much later in the text: "Preferential treatment involves the establishment of better (competitive) environment for business and investment activities as compared to similar territories [21]. Due to this, the core documents on strategic development consider the Free port of Vladivostok as an instrument of advanced economic development [12,13]."

Point 3

We added some material in part 1 to make more clear what is "preferential status" and how it relates to sustainable development

Preferential treatment in ports is a special regime that supposes lowering taxes, introducing special investment programs and strengthening of economic development

and

In this paper the authors tried to estimate the results of preferential treatment program of the free port of Vladivostok and compared it with the pioneer experience of sustainable development of the port of Antwerp.

Point 4

The following statement is not appropriate for the main text and the exchange rate should be a note to the table it refers to: "We provide the financial performance of the Russian port of Vladivostok using statistics in rubles. Euro to Russian Ruble Exchange Rate is at a current level of 89.05 (18.02.2021)". The use of 'we' should be avoided and the exchange rate made more obvious.

Point 4

We made the exchange rates obvious eacn time when rubles are mentioned in the article.

Point 5

In section 4.1, 'residents' can be interpreted as people or population, when clearly what is meant is companies or businesses. The text should always use the expression ‘resident companies’ rather than just residents, particularly since later in the text, populations (residents) are referred to.

Point 5

The term “residents” was changed to “resident companies”

Point 6

The numerical figure in the following sentences is extremely confusing to a western reader.” The total amount of privileges provided for the payment of customs duties and taxes amounted to 56,576. 62 thousand rubles.”, “The privileges provided for the payment of customs duties and taxes amounted to 42,483. 79 thousand rubles”. The figures after the decimal point are trivia´l and should be deleted and the whole number figure should be quoted as either 'million' or and extra three zeros 'i.e. 000' should suffix each of these numbers. 

Point 6

The numerical figures were changed to make them easier to understand. “42,483. 79 thousand rubles” was changed to “42.5 million rubles”. The same approach was used to all figures in text.

Point 7

Section 4.3 is structurally unsound. The section on the port of Antwerp governance and regulation is largely irrelevant to the central discussion and should be deleted. The section comparing Antwerp and Vladivostok should be moved to the beginning of section 4.4. as a form of introduction to the situation in Vladivostok in comparison to Antwerp. Section 4.3 then becomes only about Antwerp.

Point 7

The section on the port of Antwerp governance and regulation was be deleted. The section comparing Antwerp and Vladivostok was moved to the beginning of section 4.4. References and links to them were corrected.

Reviewer 2 Report

Some editing is necessary in a few points:

For example: abstract line....15 needs editing

or line 456 "..integrated coastal area management is an effective technology proven..." (ICAM is not a "technology" ...may be an approach

also it would be a good idea to analyze what is "preferential status" and how it relates to sustainable development

Author Response

Point 1

Some editing is necessary in a few points:

For example: abstract line....15 needs editing

Point 1

We edited the sentence at line 15: 

The authors analyzed the Russian Audit Chamber report on the activities of the free port of Vladivostok and compared it with the pioneer experience of sustainable development of the port of Antwerp. 

Point 2

or line 456 "..integrated coastal area management is an effective technology proven..." (ICAM is not a "technology" ...may be an approach

Point 2

The term “technology” was changed by the term “approach”

Point 3

also it would be a good idea to analyze what is "preferential status" and how it relates to sustainable development

Point 3

We added some material in part 1 to make more clear what is "preferential status" and how it relates to sustainable development

Preferential treatment in ports is a special regime that supposes lowering taxes, introducing special investment programs and strengthening of economic development

and

In this paper the authors tried to estimate the results of preferential treatment program of the free port of Vladivostok and compared it with the pioneer experience of sustainable development of the port of Antwerp.

Reviewer 3 Report

The manuscript attempts to conceptualize the Sustainable Development Goals as an element of integrated coastal preferential zone management and concludes that preferential treatment should be applied allowing problem solutions, laying the foundation for long-term sustainable development and mutually beneficial cooperation in the coastal region.

I believe that the manuscript is worth publishing and requires only minor corrections.

Ln 165 …are not used...

Ln 179-180. Reduce Table 3 font size to make it more presentable (especially column headings).

Ln 220 …we should compare…

Ln 224-226. Reduce Table 4 font size to make it more presentable (especially column headings).

Ln 234 …estimated investment was about…

Ln 242 Thus, we suggest that… (I do not see any reasoning for conclusion but more of a suggestion).

Ln 289-290. Make sure the table font and size are uniform in Table 5.

Ln 500…Vladivostok operation available from open sources.

Author Response

Point 1

Ln 165 …are not used...

Point 1

“are not use” was changed to “are not used”

Point 2

Ln 179-180. Reduce Table 3 font size to make it more presentable (especially column headings).

Point 2

font size was reduced

Point 3

Ln 220 …we should compare…

Point 3

…we should to compare… was changed to …we should compare…

Point 4

Ln 224-226. Reduce Table 4 font size to make it more presentable (especially column headings).

Point 4

font size was reduced

Point 5

Ln 234 …estimated investment was about…

Point 5

…estimated investment is about…was changed to …estimated investment was about…

Point 6

Ln 242 Thus, we suggest that… (I do not see any reasoning for conclusion but more of a suggestion).

Point 6

we conclude that… was changed to …we suggest that…

Point 7

Ln 289-290. Make sure the table font and size are uniform in Table 5.

Point 7

Table font was changed and font size was reduced

Point 8

Ln 500…Vladivostok operation available from open sources.

Point 8

Vladivostok operation than available from open sources was changed to …Vladivostok operation available from open sources.